## Perspective

climate change; collective mental health; global mental health; climate anxiety; eco-anxiety; eco-paralysis

**Corresponding author:**
Deborah Oluwaseun Shomuyiwa;
Email: deborahshomuyiwa@gmail.com

# Climate change trauma and collective dissociation: Unraveling the impact on mental health and advocating for collective action

Deborah Oluwaseun Shomuyiwa[1,2] and Don-Eliseo Lucero-Prisno III[3,4,5]

[1]Department of Health Promotion and Behavior, College of Public Health, University of Georgia, Athens, Georgia, USA; [2]Global Health Focus, Lagos, Nigeria; [3]Department of Global Health and Development, London School of Hygiene & Tropical Medicine, London, United Kingdom; [4]Faculty of Management and Development Studies, University of the Philippines Open University, Los Baños, Laguna, Philippines and [5]Faculty of Public Health, Mahidol University, Bangkok, Thailand

## Abstract

The climate change crisis is a complex global challenge that has far- reaching implications for public health and well-being. Rising temperatures and more frequent extreme weather events are impacting physical health, mental well-being, and ecological balance. Vulnerable communities are disproportionately affected, especially in terms of food security. Furthermore, climate-related disasters have profound and lasting effects on mental health, leading to trauma responses and dissociation as coping mechanisms. This perspective delves into the concept of collective dissociation, a subconscious defense mechanism that hinders effective action in the face of the overwhelming climate crisis. Understanding and characterizing this phenomenon is essential to promote meaningful climate action. To combat collective dissociation and facilitate effective collective action, several strategies are proposed. Responsible information management for advocacy, local moral support, strategic policy development, and research on climate trauma processing are highlighted as vital approaches. By addressing the mental health implications of climate change, raising awareness, and prioritizing resilience and cooperation, societies can transcend collective dissociation and work together towards a more sustainable future for both the planet and its inhabitants. This call to action underscores the need for comprehensive and guided measures to safeguard planetary and population health in the face of this pressing crisis.

## Impact statement

This article explores the critical issue of collective dissociation in the context of climate change, highlighting how psychological and relational dynamics impede effective action. By identifying collective dissociation as a major barrier to addressing the climate crisis, the study underscores its impact on public health, mental well-being, and ecological stability. It proposes targeted interventions, including responsible information management, local moral support, and strategic policy development, to combat inaction and denial. The research emphasizes the need for integrating climate action into governance and resource allocation, aiming to foster global awareness, cooperation, and community empowerment. Its findings are poised to inform policy decisions and drive societal change, promoting a shift towards proactive and empathetic climate activism with lasting implications for health systems and policies.

## Introduction

Climate change is a profound and multifaceted crisis that transcends being merely an environmental issue. It represents a complex phenomenon impacting the environment, health, human behavior, and the economy in interconnected ways (Cianconi et al., 2020). Beyond its tangible environmental effects, climate change poses significant mental health challenges, presenting both immediate and long-term implications (Ramadan and Attallah, 2021). Extreme weather events, exacerbated by global warming, can trigger a wide range of psychopathological responses, including mood disturbances, anxiety, and physical symptoms (Cianconi et al., 2020). These events also contribute to lasting mental health issues such as depression, post-traumatic stress disorder (PTSD), increased suicide rates, and substance abuse, particularly among vulnerable populations and those displaced by environmental factors (Cianconi et al., 2020; Cruz et al., 2020).

As the world grapples with the escalating impacts of climate change, a unique form of trauma, known as "climate change trauma," is emerging (Bednarek, 2021; Ramadan and Ataallah, 2021; White, 2015). Unlike traditional trauma, which is typically experienced at the individual level,

climate change trauma permeates entire communities and societies. This phenomenon leads to what is termed Collective Dissociation (Bednarek, 2021), a coping mechanism for the overwhelming global trauma inflicted by climate crisis-induced disasters (White, 2015). Addressing the full spectrum of mental health responses to climate change requires a holistic approach that integrates biological, psychological, behavioral, and social dimensions (Bednarek, 2021). This approach is essential for developing effective strategies for adaptation and resilience, as well as promoting collective action.

Moreover, the climate crisis fundamentally represents a crisis of relationships. The disruption of natural systems parallels the disruption of social and psychological systems, creating a profound relational crisis between humans and the planet, as well as within communities globally (APA, 2021). This relational dimension necessitates a response from mental health professions and the broader public health community, extending beyond merely addressing distress caused by climate change. Recognizing this, the perspective aims to highlight the significant mental health impacts of climate change, introducing the concept of climate change trauma and collective dissociation as societal responses to this crisis.

## Climate change and mental health trauma

Climate change induces widespread psychological distress and trauma, impacting individuals, communities, and societies (Ramadan & Ataallah, 2021; Woodbury, 2019). Woodbury identifies climate trauma as a missing narrative linking global inaction to the climate crisis. Massaro et al. (2018) identify two reactions: reexperiencing and dissociation. Bednarek (2021) discusses collective trauma manifesting as fragmentation, polarization, and dissociation within cultures and societies. Dissociation, a natural psychological and neurobiological self-protection mechanism, varies in presentation based on the nature of the crisis (White, 2015). When anxiety and distress translate into denial, especially 'collective dissociation,' it significantly affects the perception and conceptualization of the climate crisis.

The existential threat posed by climate change to global biodiversity and shared identity deeply impacts human minds (Li et al., 2022). As members of the biosphere, humans experience overwhelming stress from the perceived lack of protection against environmental assaults (Woodbury, 2019). Climate trauma can trigger past personal, cultural, and intergenerational traumas, leading to psychosocial defense mechanisms that cause populations to recoil from the climate crisis implications. Real-time climate trauma's impact on social structures becomes evident when its environmental marks surface (Woodbury, 2019). This growing awareness of global interconnectedness and victims' mentality induces a departure from mental reality (Massazza, Ardino, & Fioravanzo, 2022). The effects often manifest as acute stress, leading to long-term anxiety and depression, sometimes necessitating professional intervention (Li et al., 2022).

Climate change has been recognized as a stressor exacerbating PTSD and other mental health conditions (Massazza, Ardino, & Fioravanzo, 2022). Direct consequences, such as extreme heat and weather events, worsen pre-existing mental health conditions (Charlson et al., 2021). Feelings of distress, powerlessness, and hopelessness are grand-scale trauma responses (White, 2015). Heightened awareness and concern about the climate crisis correlate with increased stress and poorer mental health (Woodbury, 2019). Climate change is marked by critical thresholds or tipping points, leading to significant, often irreversible climate system

changes once crossed. Mitigating tipping point risks is crucial for effective climate policy (Cianconi et al., 2020). Recognition of the crisis's effects on victims of climate-induced trauma is growing, but the human mind tends to dismiss thoughts of trauma. Psychological defenses create aversion to trauma itself and its cumulative impacts, termed 'reflexive resistance,' diminishing acknowledgment of climate change and its importance.

Concerns about climate change mitigation and adaptation risks evoke various emotions, including anxiety, stress, and psychological discomfort. These feelings link to perceived risks—functional, physical, financial, social, or psychological—associated with climate change (Gifford, 2014; Gifford, 2011). These perceived risks contribute to climate trauma, intensifying future uncertainty and insecurity. McDonald et al.'s (2016) review highlights the impact of personal experiences, such as extreme weather events or witnessing climate pattern changes, on shaping beliefs, concerns, and motivation for climate action. However, the relationship between personal experience and climate change beliefs is complex, moderated by factors like worldviews, political ideology, and cultural values (McDonald, 2016).

## Collective dissociation and health systems

Collective dissociation, a form of trauma processing, threatens the cooperation needed to tackle climate change (White, 2015). When societies cannot process the enormity of this threat, rational aspects may continue to function while emotional complexities become fragmented (Bednarek, 2021). This fragmentation hampers integrated, adaptive responses to climate change, reinforcing isolation and preventing an objective assessment of its destructive reality. Consequently, human nature remains entrenched in harmful environmental practices, undermining planetary health even when climate action is critical (Lengieza, Aviste, & Richardson, 2023). Collective dissociation affects all levels of society, leading to social detachment and political apathy, which hinders meaningful climate action (Hornung, 2022). Vulnerable populations are particularly impacted, making it crucial for health systems to address this form of trauma to protect planetary health.

The psychological impact of climate change extends beyond being a mere victim of environmental shifts. It serves as a significant impediment to proactive climate action. The overwhelming scale and complexity of the issue often leave individuals feeling powerless, leading to a sense of futility in their ability to effect meaningful change (Wamsler & Bristow, 2022). This helplessness can manifest as sleep disturbances and heightened anxiety, exacerbating the mental toll of climate change (Dodds, 2021). Discussions about climate change often evoke existential fears, triggering defense mechanisms like denial to cope with anxiety (Dodds, 2021; Davy, 2021). Terror management theory explains that reminders of mortality cause individuals to use psychological defenses to manage existential anxiety (Davy, 2021; Myers, 2014). In the context of climate change, this leads to defensive strategies that hinder environmentalism (Myers, 2014). People might distance themselves from the reality of environmental degradation, preventing meaningful action.

However, the concept of collective dissociation is just one aspect of a complex phenomenon - the relationship between the mind and climate change is not linear. People's inertia towards climate action is influenced by psychological, social, cultural, and political dynamics (Brulle & Norgaard, 2019). Dissociation is only one of many psychological responses, including denial, fear, and

misinformation, that contribute to this inertia (Moser, 2016). These responses are shaped by broader systemic issues, such as the influence of fossil fuel industries on political decision-making (Munck af Rosenschöld, 2014; Gifford, 2014) Phenomena like tokenism and rebound effects can undermine individual actions aimed at mitigating climate change (Bednarek, 2021). When efforts are perceived as superficial or ineffective, individuals may experience frustration or disillusionment, contributing to a sense of collective dissociation or powerlessness in addressing the issue (McDonald, Chai, & Newell, 2016; Gifford, 2014). Addressing these psychological and systemic barriers is crucial for fostering effective climate action and protecting mental health.

## Acting for collective action

Protecting population health is essential to breaking the cycle of climate trauma. An integrated approach is needed to address global collective dissociation.

### *Characterizing collective dissociation*

Closing the gap of climate crisis-mediated collective inaction necessitates a thorough characterization of the varied responses to the climate crisis. Collective dissociation has been observed to influence both directly impacted and climate-informed populations. Acknowledging that collective dissociation related to climate trauma is a significant psychological barrier to addressing the climate crisis is a crucial step. Inaction and lack of engagement with the collective predicament of the climate crisis should also be recognized as important responses. Promoting healing while coordinating change, self-care, responsibility, and transformation in social structures for planetary health requires individual and social awareness of this concept. By viewing the climate crisis through the lens of trauma, we can shift climate change activism towards a more functional expression and reaction, fostering a steady but conscious awakening to empathy and collective healing. This perspective helps to humanize the issue, making it more relatable and actionable for individuals and communities. By fostering a sense of shared responsibility, ethical awareness, and community engagement, individuals and societies can move towards more effective and emotionally sustainable approaches to the climate crisis. Emphasizing the power of collective action can help mitigate the feelings of isolation and fragmentation that often accompany collective dissociation. This collective approach provides emotional support and solidarity, essential for coping with climate trauma and driving sustainable action.

### *Information management for advocacy*

Effective information management for advocacy is crucial for raising climate change awareness and fostering participation across global health systems. Advocacy narratives must balance individual responsibility with the roles of corporations, industries, and political structures. While individual actions matter, focusing solely on personal responsibility can lead to high emotions, overwhelming experiences, paralysis, and inaction (Li et al., 2022). Overemphasis on personal responsibility obscures broader systemic issues and the significant impact of corporate and political actions. Advocacy should highlight the importance of systemic change and stress the accountability of larger entities like governments and corporations. This approach prevents exonerating powerful actors and ensures they are held accountable for their substantial contributions to climate change.

Supporting vulnerable populations in addressing climate change is critical. This includes ensuring access to essential survival amenities and addressing the mental health impacts of climate change. Advocacy should promote messages of courage and proactiveness, avoiding fearmongering that can lead to survivor dissociation (Woodbury, 2019). A realistic view of the climate crisis empowers communities to take action, fostering resilience and adaptation. Emphasizing collective action and community engagement is vital. Highlighting successful examples of community-based renewable energy projects or local climate resilience initiatives can inspire broader participation and demonstrate the tangible impacts of collective action. Encouraging individuals to join forces with their communities creates a sense of solidarity and shared purpose, essential for sustaining long-term engagement. Tailoring advocacy messages to align with the dominant moral concerns of different ideological groups is necessary for garnering bipartisan support. Understanding how values, beliefs, and group norms influence climate change concern and action can inform effective communication strategies. By resonating with diverse audiences, advocacy efforts can bridge divides and foster a more inclusive approach to climate action.

Creating consciousness-raising safe public spaces is essential for affirming climate truth and promoting "respond-ability." These spaces can serve as forums for discussion, education, and support, helping individuals process their emotions and experiences related to climate change. The health community can play a significant role by advocating for mental health protection and providing resources to support emotional well-being. Education and awareness-raising are critical to ensuring that current and future generations recognize and address environmental degradation. Advocacy should include educational initiatives that highlight historical environmental conditions and the changes over time, fostering a deeper understanding of what has been lost and what can be regained. Advocacy messages should promote resilience and positive adaptation strategies. Highlighting stories of adaptation and innovation can inspire hope and action. Encouraging communities to see themselves as capable of effecting change empowers them to take proactive steps in addressing climate challenges. By focusing on collective action, tailored messaging, and education, advocacy can drive meaningful climate action and foster resilience across communities.

### *Local moral support*

Local moral support is vital for fostering sustainable adaptation strategies and community resilience to climate change. Encouraging community awareness and participation at the local level enhances capacity and mental preparedness for climate action. Building capacity within local communities involves organizing vulnerability assessments and developing tailored action plans. These plans should include measures for improving infrastructure, enhancing social cohesion, and providing mental health support, reducing vulnerability to climate-related stresses. Community engagement is key to effective local adaptation. Encouraging participation in climate action initiatives empowers individuals and fosters a sense of ownership and responsibility. Educational programs, workshops, and public forums promote climate awareness and action, emphasizing the interconnectedness of local actions and global outcomes.

Social structures play a vital role in addressing collective dissociation and promoting recovery. Empowerment and reconnection are central to this process. Local initiatives that focus on building social ties and fostering a sense of community can help mitigate the feelings of isolation and fragmentation often associated with climate trauma. Community gardens, renewable energy projects, and climate action groups provide practical ways for individuals to engage and support each other. Shared awareness is a powerful tool for reconciliation with the current narrative of protecting humanity and planetary health. Creating safe public spaces for discussing climate issues and sharing experiences is important. These spaces serve as hubs for education, support, and action, promoting a collective response to climate challenges. Support groups and networks offer vital emotional and psychological support, aiding individuals in navigating their climate-related anxieties and traumas. Fostering shared responsibility and collective action mitigates collective dissociation. Emphasizing community engagement reduces despair, providing essential solidarity for coping with climate trauma. Local leadership should support community-based climate action by providing resources and frameworks, facilitating engagement, and ensuring mental health support is accessible.

## Policy development

Policy development is pivotal in tackling the intricate challenges of climate change and its associated traumas. Effective policies drive collective action, foster cooperation, and bolster resilience across society. Policymakers must frame climate change as a global crisis, prioritizing the mental health impacts of trauma induced by climate stressors. Adopting a comprehensive, strategic approach is imperative. This involves integrating climate action across all sectors, allocating sufficient financial resources, and making mental health resilience a cornerstone of climate strategies (Brulle & Norgaard, 2019). For example, the Drought Resistance Alliance, launched at COP27, can serve as a model by prioritizing mental health resilience and providing participatory support for affected communities.

However, to make meaningful progress, it is crucial to move beyond rhetorical calls for citizen action and directly confront the systemic forces that perpetuate the crisis. Corporate resistance, government paralysis, and the prioritization of profits over environmental sustainability are key drivers of inaction. The financial sector's continued investment in coal and oil, alongside unchecked consumerism, greenwashing, and the war economy, contribute significantly to the climate emergency. These dynamics alienate citizens, creating a disconnect between the visible dangers of climate change and the lack of meaningful action by those in power (Wamsler & Bristlow, 2022). Policy development must therefore prioritize accountability for corporations and governments, recognizing that citizen-driven initiatives alone will not suffice in the face of such powerful opposition (Wamsler & Bristlow, 2022). Understanding the socio-political drivers of climate inaction, including the influence of political contributions from the fossil fuel industry, is crucial. Policymakers must strive for greater transparency and accountability in political funding to align actions with public support for effective climate measures.

Policies should also emphasize education and public awareness campaigns to raise consciousness about the mental health consequences of climate change. By fostering a deeper understanding of climate trauma, these campaigns can mobilize public support for climate action and promote collective healing and resilience. Education initiatives should focus on overcoming environmental generational amnesia, where successive generations accept degraded

environmental conditions as normal. This can be achieved through curriculum development, community programs, and media outreach. A long-term vision is crucial for the success of climate policies. Policymakers should prioritize sustainable practices, accountability, monitoring progress rigorously, and remaining adaptable to research and technological advancements. By continuously evolving and adapting policies, governments can ensure they are effectively addressing both the immediate and long-term impacts of climate change. Policies should prioritize the needs of vulnerable populations and aim to build inclusive, resilient communities.

Education and public awareness campaigns must spotlight the mental health repercussions of climate change. By enhancing understanding of climate trauma, these campaigns can mobilize public support for climate action and promote collective healing and resilience. Initiatives should tackle environmental generational amnesia, where degraded conditions are normalized over generations. Effective policies should prioritize sustainability, accountability, and adaptability to research and technological advancements. The continuous evolution of policies is necessary to address both immediate and long-term climate impacts, ensuring that the needs of vulnerable populations are prioritized and resilient, inclusive communities are built.

## Research

The growing field of research on climate trauma and collective dissociation provides a crucial foundation for understanding the sociological and psychological dimensions of climate change. This research not only informs interventions to improve mental health and well-being in affected populations but also explores the interrelationships between psychological distance, denial, and resilience. Comprehensive studies are needed to explore these dynamics and their effects on climate behaviors, aiding in the design of communication strategies that resonate with diverse audiences and bolster public engagement and action.

Research should also investigate the moderating effects of ideology on responses to climate change. Understanding how different ideological perspectives influence perceptions and reactions can inform message tailoring for bipartisan support, bridging political and social divides. Collaboration among psychologists, therapists, scientists, ecologists, and activists is vital for developing interventions that address mental health impacts and broader social implications. Interdisciplinary research can explore the effectiveness of therapeutic practices in alleviating climate anxiety and promoting resilience.

Examining historical responses to climate change offers valuable lessons for contemporary efforts. By learning from past successes and failures, researchers can identify effective strategies for adaptation and resilience. Prioritizing the mental health implications of climate change is crucial. Studies should focus on the psychological effects of climate-related stressors and how collective dissociation hinders action. Understanding these barriers can guide the development of interventions that promote mental well-being and engagement with climate issues.

Cross-disciplinary collaboration will be key to advancing therapeutic practices that alleviate climate anxiety and foster a collective healing process. These efforts will contribute to uniting humanity in the face of the climate crisis, emphasizing the need for a holistic approach that integrates mental health, environmental sustainability, and social justice. Bringing together experts from various fields can lead to a more integrated understanding of climate trauma and

dissociation, driving the creation of comprehensive policies and interventions that address the multifaceted nature of climate change impacts.

## Conclusion

Collective dissociation hinders effective climate action. To create a safer climate environment, we must address the mental health impacts of climate change and understand the collective trauma response. A shared lack of awareness diminishes the transformative power of collective action, posing a dangerous threat to climate initiatives. This is a compelling call to action to proactively address these mental health implications. By leveraging research, advocacy, policy development, and collaboration, we can mitigate the impact of collective dissociation on health systems and strive for a more resilient and sustainable future.

**Open peer review.** To view the open peer review materials for this article, please visit http://doi.org/10.1017/gmh.2024.119.

**Data availability statement.** Data availability is not applicable to this article as no new data were created or analyzed in this study.

**Acknowledgements.** Not Applicable.

**Author contribution.** Deborah Oluwaseun Shomuyiwa: Conceptualization, Project administration, Writing – original draft, Writing – review & editing; Don Eliseo Lucero-Prisno III: Conceptualization, Supervision, Writing – review & editing.

**Financial support.** This research received no specific grant from any funding agency, commercial or not-for-profit sectors.

**Competing interest.** The authors declare no conflict of interest.

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
