## [Reviewer Report]

Article title: Change Trauma and Collective Dissociation:

Unravelling the Impact on Mental Health and Advocating for

Collective Action

In general:

The article doesn’t say anything particularly new or unknown.

The concept of dissociation seems to be very limited, describing all mental states related to inertia towards climate change. It does not explain people’s general inertia towards climate change. This is a complex and multifaceted phenomenon. People’s distance from the climate crisis is also the product of psychological phenomena (e.g. psychological distance), political interests, misinformation, fear, which are already well explained in the literature and are well known.

We suggest you read more than White, Benjamin. 2015. Otherwise the article seems rhetorical.

The topic chosen by the authors is admirable, but the issue is much more complex than described in the text of the article.

The authors have read little about why people do not respond to climate change.

Their bibliography is limited.

Specific:

Page 6 (31 - 33) "mindset inadvertently promotes climate change (White, 2015), leading to distinct local reactions, miscommunication, trends, and disconnection on a larger scale

Comment: There is no reason to call the many phenomena of inactive response to climate change “collective dissociation”. People go through different phenomena that explain their reactions; these phenomena are also cultural, social and therefore psychological.

I recommend reading (Cianconi, P.; Hanife, B.; Grillo, F.; Zhang, K.; Janiri, L. Human Responses and Adaptation in a Changing Climate: A Framework Integrating Biological, Psychological, and Behavioural Aspects. Life 2021, 11, 895. https://doi.org/10.3390/life11090895 )

Page 6 (38 - 40) This nonconscious and automatic response leads to social detachment and political apathy, hindering progress toward meaningful climate action.

Page 7 (23-27). Collective dissociation has been

observed to influence both directly impacted and climate-informed populations. Acknowledging that collective dissociation related to climate trauma is a significant psychological barrier to addressing the climate crisis is a crucial step.

Comment: Isn’t it better to write subconsciously rather than unconsciously?

There are also fully conscious phenomena that are active in dissociating themselves from the climate crisis and disinterested in it. Why aren’t they mentioned? There is a danger of oversimplification.

I recommend reading: (Moser, S.C. Communicating climate change: History, challenges, process and future directions. Wiley Interdisc. Rev. Clim. Chang. 2010, 1, 31–53) - Moser, S.C. Reflections on climate change communication research and practice in the second decade of the 21st century: What more is there to say? Wiley Interdisc. Rev. Clim. Chang. 2016, 7, 345–369.) - (Norgaard KM. Living in Denial: Climate Change, Emotions, and Everyday Life. Cambridge, MA: MIT Press; 2011.), (McDonald RI, Chai HY, Newell BR. Personal experience and the ‘psychological distance’ of climate change: an integrative review. J Environ Psychol 2016, 44:109–118)

(Milfont, T. L. (2010). Global warming, climate change and human psychology. Psychological approaches to sustainability: Current trends in theory, research and practice, 19, 42.) - (Gifford R. The dragons of inaction: psychological barriers that limit climate change mitigation and adaptation. Am Psychol 2011, 66:290–302) and finally: (Pawlik K. (1991) The Psychology of Global Environmental Change: Some Basic Data and an Agenda for Cooperative International Research. International Journalof psychology 26, 547-563)

---

## [Reviewer Report]

Let me preface my comments by saying that this a most welcome contribution to the field of climate psychology, one that has the potential to bridge the disconnect between public opinion and political action. I’m grateful to the authors for this insightful analysis, and look forward to its publication and broad dissemination.

“Climate change can have various mental health impacts...” This reinforces the split b/t humans and nature by not acknowledging the symbiosis of life on a living planet. What is “climate change” but a measurement of variations in temperature; i.e., a scientific term based on objectifying nature. What happens if, instead, you posit “Climate trauma can manifest in many ways on a living planet where everything is interconnected.” ? Here, the subject “climate trauma” refers not to some scientific measurements, but rather to the real problem - a living planet that is experiencing a potentially lethal assault on its life-support system. This is an e.g. of how we are trying to think our way out of a problem with the same thinking that got us into the problem, no? “Understanding the full spectrum of mental health responses and challenges to climate trauma is crucial.” See how differently that lands? Dissociation becomes a coping mechanism for overwhelm from collective, global trauma, rather than a response to a larger variation in scientific measurements, which is simply not what is happening here. Leave the science to the scientists! We don’t have to accept the scientific-materialist narrative that still largely denies Gaia theory, in spite of the fact that it is no longer a hypothesis. In reality, it is the scientific-materialist objectivity that has been disproven, and so it is a grave error to continue treating Gaia as a giant test tube, as well as using language that sees humans as separate from, and thus affected by, this larger living organism that is dying from the effects of our affliction.

w/re: reflective resistance, I think it is important to point out that social taboos also play an important role in reinforcing those psycho-defenses. There is still a pretty big taboo against speaking about trauma in social context, which often lacks the kind of safety people feel in more intimate settings where trauma is more likely to be acknowledged.

p. 6, line 47: global POLITICAL resistance. Don’t blame people for the sins of their representatives. In the U.S., well over 70% of the populace has for some time now supported more drastic actions on the climate then they are ever offered by the politicians running for or holding political office. I think it quite sloppy to conflate these two camps. It is in politics where the collective dissociation is most pronounced, not in the polity. My hope is that your paper ends up catalyzing a political intervention, but you need to be much more clear here in pointing out the disconnect b/t popular support for climate action and politicians who take large contributions from the fossil fuel industry. In truth, the politicians knowingly enable and reinforce the very collective dissociation you are targeting here.

This presumption that it is essential to avoid over-emphasizing individual responsibility misses the mark by a wide margin. Please consult Thompson, A. (2012) “The Virtue of Responsibility for the Global Climate” on this point. (I wrote about this topic, citing Prof. Thompson’s work, here: https://www.whatisemerging.com/opinions/what-if-we-are-all-we-ve-got) This is also relevant to the next paragraph at the top of p. 8. And on that point, it wouldn’t hurt to cite a couple of good e.g.’s, such as Joanna Macy’s “The Work that Reconnects” “The Good Grief Network” and “The Climate Psychology Alliance”

On the topic of local moral support, please consult the APA’s “Action Plan for Psychologists” (Feb. 2022), which speaks to the moral responsibility of mental health professionals to be more proactive in their communities. chrome-extension://efaidnbmnnnibpcajpcglclefindmkaj/https://www.apa.org/science/about/publications/climate-crisis-action-plan.pdf

Excellent work! Thank you so much for allowing me the opportunity to review this important paper.

---

## [Editor Report]

Article title: Change Trauma and Collective Dissociation:

Unravelling the Impact on Mental Health and Advocating for

Collective Action

Revisor 1

In general: The article doesn’t say anything particularly new or unknown.

The concept of dissociation seems to be very limited, describing all mental states related to inertia towards climate change. It does not explain people’s general inertia towards climate change. This is a complex and multifaceted phenomenon. People’s distance from the climate crisis is also the product of psychological phenomena (e.g. psychological distance), political interests, misinformation, fear, which are already well explained in the literature and are well known.

We suggest you read more than White, Benjamin. 2015. Otherwise the article seems rhetorical.

The topic chosen by the authors is admirable, but the issue is much more complex than described in the text of the article.

The authors have read little about why people do not respond to climate change.

Their bibliography is limited.

Specific:

Page 6 (31 - 33) "mindset inadvertently promotes climate change (White, 2015), leading to distinct local reactions, miscommunication, trends, and disconnection on a larger scale

Comment: There is no reason to call the many phenomena of inactive response to climate change “collective dissociation”. People go through different phenomena that explain their reactions; these phenomena are also cultural, social and therefore psychological.

I recommend reading (Cianconi, P.; Hanife, B.; Grillo, F.; Zhang, K.; Janiri, L. Human Responses and Adaptation in a Changing Climate: A Framework Integrating Biological, Psychological, and Behavioural Aspects. Life 2021, 11, 895. https://doi.org/10.3390/life11090895 )

Page 6 (38 - 40) This nonconscious and automatic response leads to social detachment and political apathy, hindering progress toward meaningful climate action.

Page 7 (23-27). Collective dissociation has been

observed to influence both directly impacted and climate-informed populations. Acknowledging that collective dissociation related to climate trauma is a significant psychological barrier to addressing the climate crisis is a crucial step.

Comment: Isn’t it better to write subconsciously rather than unconsciously?

There are also fully conscious phenomena that are active in dissociating themselves from the climate crisis and disinterested in it. Why aren’t they mentioned? There is a danger of oversimplification.

I recommend reading: (Moser, S.C. Communicating climate change: History, challenges, process and future directions. Wiley Interdisc. Rev. Clim. Chang. 2010, 1, 31–53) - Moser, S.C. Reflections on climate change communication research and practice in the second decade of the 21st century: What more is there to say? Wiley Interdisc. Rev. Clim. Chang. 2016, 7, 345–369.) - (Norgaard KM. Living in Denial: Climate Change, Emotions, and Everyday Life. Cambridge, MA: MIT Press; 2011.), (McDonald RI, Chai HY, Newell BR. Personal experience and the ‘psychological distance’ of climate change: an integrative review. J Environ Psychol 2016, 44:109–118)

(Milfont, T. L. (2010). Global warming, climate change and human psychology. Psychological approaches to sustainability: Current trends in theory, research and practice, 19, 42.) - (Gifford R. The dragons of inaction: psychological barriers that limit climate change mitigation and adaptation. Am Psychol 2011, 66:290–302) and finally: (Pawlik K. (1991) The Psychology of Global Environmental Change: Some Basic Data and an Agenda for Cooperative International Research. International Journalof psychology 26, 547-563)

Revisor 2

Let me preface my comments by saying that this a most welcome contribution to the field of climate psychology, one that has the potential to bridge the disconnect between public opinion and political action. I’m grateful to the authors for this insightful analysis, and look forward to its publication and broad dissemination.

“Climate change can have various mental health impacts...” This reinforces the split b/t humans and nature by not acknowledging the symbiosis of life on a living planet. What is “climate change” but a measurement of variations in temperature; i.e., a scientific term based on objectifying nature. What happens if, instead, you posit “Climate trauma can manifest in many ways on a living planet where everything is interconnected.” ? Here, the subject “climate trauma” refers not to some scientific measurements, but rather to the real problem - a living planet that is experiencing a potentially lethal assault on its life-support system. This is an e.g. of how we are trying to think our way out of a problem with the same thinking that got us into the problem, no? “Understanding the full spectrum of mental health responses and challenges to climate trauma is crucial.” See how differently that lands? Dissociation becomes a coping mechanism for overwhelm from collective, global trauma, rather than a response to a larger variation in scientific measurements, which is simply not what is happening here. Leave the science to the scientists! We don’t have to accept the scientific-materialist narrative that still largely denies Gaia theory, in spite of the fact that it is no longer a hypothesis. In reality, it is the scientific-materialist objectivity that has been disproven, and so it is a grave error to continue treating Gaia as a giant test tube, as well as using language that sees humans as separate from, and thus affected by, this larger living organism that is dying from the effects of our affliction.

w/re: reflective resistance, I think it is important to point out that social taboos also play an important role in reinforcing those psycho-defenses. There is still a pretty big taboo against speaking about trauma in social context, which often lacks the kind of safety people feel in more intimate settings where trauma is more likely to be acknowledged.

p. 6, line 47: global POLITICAL resistance. Don’t blame people for the sins of their representatives. In the U.S., well over 70% of the populace has for some time now supported more drastic actions on the climate then they are ever offered by the politicians running for or holding political office. I think it quite sloppy to conflate these two camps. It is in politics where the collective dissociation is most pronounced, not in the polity. My hope is that your paper ends up catalyzing a political intervention, but you need to be much more clear here in pointing out the disconnect b/t popular support for climate action and politicians who take large contributions from the fossil fuel industry. In truth, the politicians knowingly enable and reinforce the very collective dissociation you are targeting here.

This presumption that it is essential to avoid over-emphasizing individual responsibility misses the mark by a wide margin. Please consult Thompson, A. (2012) “The Virtue of Responsibility for the Global Climate” on this point. (I wrote about this topic, citing Prof. Thompson’s work, here: https://www.whatisemerging.com/opinions/what-if-we-are-all-we-ve-got) This is also relevant to the next paragraph at the top of p. 8. And on that point, it wouldn’t hurt to cite a couple of good e.g.’s, such as Joanna Macy’s “The Work that Reconnects” “The Good Grief Network” and “The Climate Psychology Alliance”

On the topic of local moral support, please consult the APA’s “Action Plan for Psychologists” (Feb. 2022), which speaks to the moral responsibility of mental health professionals to be more proactive in their communities. chrome-extension://efaidnbmnnnibpcajpcglclefindmkaj/https://www.apa.org/science/about/publications/climate-crisis-action-plan.pdf

Excellent work! Thank you so much for allowing me the opportunity to review this important paper.

---

## [Reviewer Report]

This is amazing synthesis of the surveyed articles - THANK YOU! It’s a critical message that needs to be widely distributed, as it cuts right to the heart of our inaction in the face of an existential threat. I have only two constructive comments/suggestions:

1. While it is implicit throughout, I think it very important to make it clear up front that the climate crisis is a crisis of relationship, and thus the mental health professions not only have a duty to respond to the distress it causes, but also to address the relational aspects of the climate crisis. That kind of response has been sorely lacking from institutional arms of the psychology profession in particular, and so consistent with the theme of the paper, you may wish to acknowledge the institutional dissociation of the psychology profession itself (as opposed to, e.g., Climate Psychology Alliance of U.K. & N.A.)

2. Along those same lines, I believe this paper would benefit greatly from a citation to the 2nd Report of the APA Climate Task Force: Addressing the Climate Crisis

An Action Plan for Psychologists (https://www.apa.org/science/about/publications/climate-crisis-action-plan.pdf)

---

## [Reviewer Report]

This article identifies ‘collective dissociation’ as one of the reasons why climate change is failing to achieve the outcome of bringing collective decisions together to create the right awareness and shared responsibility for effective collective action. They emphasise that understanding this dissociation is paramount to promoting meaningful climate action and breaking through inaction, denial and disconnection. The authors propose a number of strategies to promote global awareness of climate change, foster and develop resilience and cooperation, so that societies can overcome collective dissociation and work together towards a more sustainable future for the planet and its inhabitants.

The strategies were grouped into: responsible information management, local moral support, strategic policy development and research on climate trauma processing, empowering communities, policy makers and stakeholders to take decisive action, and promoting information and education.

The article is interesting and even original in some parts, but in others it can lapse into rhetorical discourse. The interesting part is the choice of the ‘collective dissociation’ mechanism. As explained in the article, there are some inaccuracies on this issue. The article introduces the concept of collective dissociation, but does not explain it correctly. Collective dissociation is certainly a new phenomenon reported in relation to climate change and needs to be studied further. However, in the past, similar phenomena have been part of ‘mass understanding and behaviour’ in the face of dark dangers, where choices are lost in creeping situations. Collective dissociation is not just an unconscious mecenism, but a set of psychological phenomena. It is a set of psychological phenomena that can be observed at the group and collective level, down to the population segment. We have published something about this in an article on human adaptation to climate change (Cianconi P, Hanife B, Grillo F, Zhang K, Janiri L. Human Responses and Adaptation in a Changing Climate: A Framework Integrating Biological, Psychological, and Behavioural Aspects. Life. 2021; 11(9):895. https://doi.org/10.3390/life11090895) Might be worth a look.

The result of such phenomena is the paralysis of people in fear of making decisions, total inertia, denial and other psychological, social, cultural dynamics. The authors rightly point out that there are other conditions attached to collective dissociation, including denial, fear and misinformation, which contribute to this inertia.

In “(policy development)” the article becomes rhetorical (like many other articles at this point). There is no mention of the real responsibilities for the climate situation: the resistance of the companies, the delaying tactics for the sake of continued profits, the paralysis of the states and the complicit governments, and all the social and global actors that increase climate change (bank and company investments in coal, unbridled consumerism, green washing, war economy, pollution, etc.). We cannot leave everything to the will of the citizens if this is not the case. Citizens are alienated because the danger they see and hear about does not match the actions of those in power. The danger does not receive the attention it deserves in the face of the interests of the elites, the financial sector, coal and oil companies, deforestation, etc. About this topic there are lot of research -

In ‘(Research)’ section, on the other hand, is interesting and fully supported. Research on climate trauma and collective dissociation is essential for understanding the sociological and psychological dimensions of climate change. To inform interventions to improve the mental health and well-being of affected populations, to explore the interrelationships between the different dimensions of psychological distance, to understand the dynamics and barriers, to design communication strategies. Finally, to explore the efficacy of therapeutic practices to alleviate climate anxiety and promote resilience, collaboration between disciplines, and research that contributes to a collective healing process that unites humanity in the face of the climate crisis.

Something about references is not written correctly:

It repeats: Cianconi, P., Betrò, S., & Janiri, L. (2020). The impact of climate change on mental health: A systematic descriptive review. Frontiers of Psychiatry, 11, 74.

https://doi.org/10.3389/fpsyt.2020.00074Cianconi, P., Betrò, S., & Janiri, L. (2020). The impact of climate change on mental health: A systematic descriptive review. Frontiers of Psychiatry,

11, 74. https://doi.org/10.3389/fpsyt.2020.0007.

Cited but not referenced: Dr Moser is cited but not included in the references (Moser, 2016).

---

## [Editor Report]

Dear authors, thank you very much for submitting and responding to comments, it is a great improvement and the reviewers forward only minor comments to us for review.

Review 1:

This is amazing synthesis of the surveyed articles - THANK YOU! It’s a critical message that needs to be widely distributed, as it cuts right to the heart of our inaction in the face of an existential threat. I have only two constructive comments/suggestions:

1. While it is implicit throughout, I think it very important to make it clear up front that the climate crisis is a crisis of relationship, and thus the mental health professions not only have a duty to respond to the distress it causes, but also to address the relational aspects of the climate crisis. That kind of response has been sorely lacking from institutional arms of the psychology profession in particular, and so consistent with the theme of the paper, you may wish to acknowledge the institutional dissociation of the psychology profession itself (as opposed to, e.g., Climate Psychology Alliance of U.K. & N.A.)

2. Along those same lines, I believe this paper would benefit greatly from a citation to the 2nd Report of the APA Climate Task Force: Addressing the Climate Crisis

An Action Plan for Psychologists (https://www.apa.org/science/about/publications/climate-crisis-action-plan.pdf)

Review 2:

This article identifies ‘collective dissociation’ as one of the reasons why climate change is failing to achieve the outcome of bringing collective decisions together to create the right awareness and shared responsibility for effective collective action. They emphasise that understanding this dissociation is paramount to promoting meaningful climate action and breaking through inaction, denial and disconnection. The authors propose a number of strategies to promote global awareness of climate change, foster and develop resilience and cooperation, so that societies can overcome collective dissociation and work together towards a more sustainable future for the planet and its inhabitants.

The strategies were grouped into: responsible information management, local moral support, strategic policy development and research on climate trauma processing, empowering communities, policy makers and stakeholders to take decisive action, and promoting information and education.

The article is interesting and even original in some parts, but in others it can lapse into rhetorical discourse. The interesting part is the choice of the ‘collective dissociation’ mechanism. As explained in the article, there are some inaccuracies on this issue. The article introduces the concept of collective dissociation, but does not explain it correctly. Collective dissociation is certainly a new phenomenon reported in relation to climate change and needs to be studied further. However, in the past, similar phenomena have been part of ‘mass understanding and behaviour’ in the face of dark dangers, where choices are lost in creeping situations. Collective dissociation is not just an unconscious mecenism, but a set of psychological phenomena. It is a set of psychological phenomena that can be observed at the group and collective level, down to the population segment. We have published something about this in an article on human adaptation to climate change (Cianconi P, Hanife B, Grillo F, Zhang K, Janiri L. Human Responses and Adaptation in a Changing Climate: A Framework Integrating Biological, Psychological, and Behavioural Aspects. Life. 2021; 11(9):895. https://doi.org/10.3390/life11090895) Might be worth a look.

The result of such phenomena is the paralysis of people in fear of making decisions, total inertia, denial and other psychological, social, cultural dynamics. The authors rightly point out that there are other conditions attached to collective dissociation, including denial, fear and misinformation, which contribute to this inertia.

In “(policy development)” the article becomes rhetorical (like many other articles at this point). There is no mention of the real responsibilities for the climate situation: the resistance of the companies, the delaying tactics for the sake of continued profits, the paralysis of the states and the complicit governments, and all the social and global actors that increase climate change (bank and company investments in coal, unbridled consumerism, green washing, war economy, pollution, etc.). We cannot leave everything to the will of the citizens if this is not the case. Citizens are alienated because the danger they see and hear about does not match the actions of those in power. The danger does not receive the attention it deserves in the face of the interests of the elites, the financial sector, coal and oil companies, deforestation, etc. About this topic there are lot of research -

In ‘(Research)’ section, on the other hand, is interesting and fully supported. Research on climate trauma and collective dissociation is essential for understanding the sociological and psychological dimensions of climate change. To inform interventions to improve the mental health and well-being of affected populations, to explore the interrelationships between the different dimensions of psychological distance, to understand the dynamics and barriers, to design communication strategies. Finally, to explore the efficacy of therapeutic practices to alleviate climate anxiety and promote resilience, collaboration between disciplines, and research that contributes to a collective healing process that unites humanity in the face of the climate crisis.

Something about references is not written correctly:

It repeats: Cianconi, P., Betrò, S., & Janiri, L. (2020). The impact of climate change on mental health: A systematic descriptive review. Frontiers of Psychiatry, 11, 74.

https://doi.org/10.3389/fpsyt.2020.00074Cianconi, P., Betrò, S., & Janiri, L. (2020). The impact of climate change on mental health: A systematic descriptive review. Frontiers of Psychiatry,

11, 74. https://doi.org/10.3389/fpsyt.2020.0007.

Cited but not referenced: Dr Moser is cited but not included in the references (Moser, 2016).

---

## [Reviewer Report]

Very interesting article indeed.

The subject focuses on a great dilemma of our time: paralisis in the face of the trend towards the collapse of societies.